# Impact of the TEI Peer Tutoring Program on Coexistence, Bullying and Cyberbullying in Spanish Schools

**DOI:** 10.3390/ijerph20196818

**Published:** 2023-09-24

**Authors:** Vanesa Sainz, O’Hara Soto-García, Juan Calmaestra, Antonio Maldonado

**Affiliations:** 1Facultad de Educación y Psicología, Universidad Francisco de Vitoria, Pozuelo de Alarcón, 29223 Madrid, Spain; o.soto@ufv.es; 2Departamento de Psicología, Universidad de Córdoba, 14071 Córdoba, Spain; m02cavij@uco.es; 3Facultad de Formación de Profesorado y Educación, Universidad Autónoma de Madrid, 28049 Madrid, Spain; antonio.maldonado@uam.es

**Keywords:** bullying, cyberbullying, coexistence, school climate, TEI program

## Abstract

The TEI peer tutoring program (in Spanish, *Tutoría entre Iguales*, hereinafter TEI ) is Spain’s most important coexistence program for the prevention of violence and bullying in secondary schools and one of the first worldwide. So that we may better appreciate the effectiveness and benefits of this program, a comparative study has been developed between four schools that are presently following this preventive strategy (TEI centres) and four other schools that do not carry out the TEI program (non-TEI centres). Controlling the other sociodemographic variables, students’ perception of coexistence, bullying and cyberbullying has been evaluated. In total, 1015 secondary school students belonging to eight schools from four autonomous communities in Spain participated. The results reveal that the students of TEI centres have a more favourable perception of educational coexistence and indicate lower rates of bullying and cyberbullying than those studying at non-TEI centres. These results highlight the benefits of the TEI program and the need to continue promoting and expanding these preventing bullying strategies in schools.

## 1. Introduction

### 1.1. Educational Coexistence

In recent decades, school coexistence has been the cause of growing concern in the educational world, mainly due to the rise in conflicting situations such as bullying and cyberbullying. These unfortunate events have generated in teachers, families and professionals a special awareness and concern. As a consequence, specific programs have been implemented in schools with subsequent detailed analyses looking into the various variables of the causes and consequences of this phenomenon [1,2,3].

The first relevant studies emerged during the 1990s, with the publication of the Delors Report, proposed by UNESCO, which establishes as one of the four fundamental pillars of education, “Learning to live together, learning to live with others” [4] (p. 6). It recognises the importance of fostering school coexistence in order to achieve a quality education and developing social skills in students that allow them to resolve conflicts, participate in common projects and respect diversity. Since then, the study of school coexistence has spread to many countries in Europe and America, giving rise to different research projects focussing on schools [5].

However, there is currently no single definition of the concept of coexistence, resulting in confusion in the absence of a clear approach to help guide research. In English, the term “coexistence” refers to the positive and proactive interaction between the different educational agents in schools [6]. 

Carbajal [7] includes in her definition the term equity, in terms of equal opportunities. This author argues that schools with a greater social and academic gap tend to report higher levels of school violence. It therefore follows that the promotion of equity is essential for the achievement of better coexistence in the classroom. However, all these authors concur that the term “coexistence” is extremely broad and highly complex, both theoretically and practically, as a result of its great dynamism [8].

In line with UNESCO’s recommendations, Hirmas and Carranza [9] underline three main dimensions that must be taken into account when considering school coexistence: inclusion, democracy and peace. The first of these concerns diversity, equal opportunities and collaboration. Democracy is related to participation in school life, conflict resolution based on dialogue, compliance with rules and respect for values. Finally, peaceful coexistence guarantees the common good, developing trust in others and in the school [10]. Other authors [11] establish integral education, student autonomy, coherence between discourses and practices, the appropriate approach to conflicts, the family–school relationship, and the care of the teaching team as key factors for school coexistence.

For their part, Fierro-Evans and Carbajal-Padilla [5] propose an indicator of the climate of school coexistence by assessing three dimensions: an environment of respect (respectful treatment among all members of the community), a structured environment (existence of clear rules and limits, known, demanded and respected by all) and a safe environment (degree of prevention and security against physical, psychological or verbal violence).

On the contrary, a dysfunctional coexistence in schools can give rise to psychosocial problems in students, worse school performance, increased relational problems between family and school and loss of student self-esteem [12,13]. Therefore, it is essential for all educational agents to actively foster coexistence and the school environment, with the aim of minimizing these consequences.

### 1.2. Bullying and Cyberbullying in Schools

A highly relevant factor that negatively affects school coexistence is the emergence of violence [14]. This phenomenon can be defined as the use of any physical, psychological, or social strategy intended to cause harm to another individual [15]. In the educational context, there are numerous types of violence, two of them being particularly serious: bullying and cyberbullying [16].

The term bullying was first defined by Olweus [17] as a type of interpersonal violence exercised between students, characterized by the intentionality of harming another defenceless student, the inequality of power between victim and aggressor, the repeated execution of the aggression and the creation of a dyadic relationship of dominance and submission between the students involved. Violence can be committed in a variety of ways, including all types of physical, verbal, psychological, emotional and social abuse of the victim. The most common physical aggressions are kicks, shoves, blows and beatings. As for verbal violence, insults, humiliations, or the use of derogatory names to call the victim are the most common offences. Other forms of abuse, such as social isolation, rejection, the spreading of rumours and marginalization are also included [18]. 

At present, the great advance of technologies has made it easier for students to access new forms of communication that are much more immediate and effective. Despite the many benefits that these advances bring, the misuse of electronic devices can lead to the emergence of other forms of violence [2]. Recent research has revealed a steady growth in a type of bullying conducted through electronic means among minors, which is referred to as cyberbullying [19]. This type of harassment presents similar characteristics to more traditional bullying, but with the difference that it occurs in digital contexts such as social networks, mobile phones, or even online games. This type of violence gives the aggressor greater control over the victim thanks to easy access to technology, the greater scope of dissemination and the possibility of remaining anonymous. In addition, cyberbullying can occur at any time and place, reaching a large number of viewers without the need to establish physical contact [20,21]. As a result, the victim often feels helpless and anxious, as sometimes they do not know the identity of their aggressor and do not have the necessary knowledge to be able to act when faced with this harassment [15].

Some manifestations of cyberbullying can be the publication of private images, the creation of false profiles, impersonation, spreading defamations of the victim, stalking, password theft, persecution on social networks, violation of privacy, exclusion and provocation. At present, these actions have legal consequences under criminal law, both for cyberbullying perpetrators and witnesses [15].

With the emergence of the COVID-19 pandemic, numerous studies have revealed an exponential growth in cyberbullying cases and, as children increasingly access electronic devices earlier, a marked increase in younger students being involved [21,22].

In relation to the roles that comprise a situation of bullying or cyberbullying, we can distinguish the roles of victim and aggressor, although there are also witnesses or spectators who can be neutral or be in favour of the aggressors or the victims [23]. All of them participate directly or indirectly in the situation of harassment, even though they may not be aware of the consequences of their actions [24]. Furthermore, it has been observed that there is a relationship, and sometimes overlap and continuity, between the profiles of victims and bullies, especially in the online context. Victims and cyber-victims often choose to respond to their aggressors online, thus also becoming cyber-aggressors [25].

### 1.3. The TEI Program for the Prevention of School Bullying

There are numerous prevention and intervention programs and projects against this type of violence, focused on different areas which are implemented in schools [26,27]. However, there is limited scientific evidence regarding the validity of these school bullying prevention programs [28,29], posing new challenges and opportunities in the development of these programs [30]. Some of the most internationally prominent are the Olweus Bullying Prevention Program [17], Pikas [31], ABC Program [32], Kiva [33], Be-Prox [34], Additionally, and Tutoring Program Between Peers-TEI, [35].

The TEI program stands out as being the most implemented in Spain, although it has also been implemented in other countries, with a large amount of subsequent scientific research to measure its effects [36]. It is a prevention program through peer-to-peer emotional tutoring aimed at improving coexistence and preventing school violence and cyberbullying in schools. The program involves the entire educational community, being essential to the collaboration and commitment of students, teachers, non-teaching staff and families [36,37].

As described by the author of the program [35], the TEI program is based on three main theoretical pillars: the theory of ecological systems by Urie Bronfenbrenner [38], the emotional education of Salovey and Mayer together with Goleman [39,40], and the positive psychology of Seligman [41], through the reproduction of positive and helpful models. The first of the pillars refers to the influence that the environment has on the cognitive, moral and relational development of the human being. The second focuses on students’ skills to understand, use and manage their own emotions and those of others, especially for conflict resolution. The latter seeks to guide education toward happiness and optimism, placing the student at the centre of the learning process. It has been demonstrated that the development of socioemotional skills contributes to the prevention of bullying and cyberbullying in schools [42,43]. The main objectives of this program are to raise awareness and make the entire educational community more alert to bullying and cyberbullying, as well as facilitate integration, create references from older tutors to provide security, empower students, compensate for the differences in power between the victim and the bully and develop a network of TEI centres that can share experiences and good practices [44].

For the development of the TEI program in educational centres, students are assigned a tutor who is two years above them at the same school. In the case of secondary education, 3rd-grade students are 1st-grade tutors, and 4th-grade students are 2nd-grade tutors. The tutors always take part voluntarily and are offered training in awareness, empathy, empowerment and individual and group commitment to promote the prevention of bullying in the school and help the younger students seeking help suffering from cases of violence [44]. This program focuses on the training of students so that they themselves are the ones who resolve conflicts in an appropriate way, avoiding violent behaviour. The established mentoring model focuses on developing cognitive, psychological, and, above all, emotional competences in students related to cooperation and teamwork. The TEI program “seeks to foster satisfactory peer relationships in which respect and tolerance prevail, encouraging the development of resilient tools for conflict resolution among students, these being the main elements of the program’s approach” [45] (p. 77).

The TEI program uses, as a practical basis for conflict resolution, a double triangle of intervention, in which the victim of bullying informs their tutor, and together, they seek a solution through dialogue. When the student-tutor fails to resolve the conflict, the victim’s tutor talks to the tutor of the student who has caused the violent situation, seeking a solution to the problem together. In cases where a solution cannot be found, the tutors ask the program coordinator (teacher) to help resolve the situation [35].

Regarding the sequencing of the implementation process, the program consists of six distinct phases: sensitization, approval, training, development, evaluation, and improvement proposals. First, the educational centre is contacted in order to raise awareness of the importance of the program. Subsequently, the educational project is approved by the school´s management team. Thirdly, teachers, students and families receive training on bullying and its consequences, are presented with the project, and are encouraged to actively take part. Students who voluntarily wish to become tutors receive more specific training on the approaches and actions to be developed, with an accreditation card being issued upon completion of the training sessions. In the fourth phase, the tutors are assigned to the younger students and activities are carried out to foster a link between them and ensure the good use of formal and informal tutorials. In the fifth phase, at the end of each term, the teaching staff and students evaluate the effectiveness of the program. Finally, in the sixth phase, an annual final report is made, and proposals for improvement are evaluated [35,36]. 

Previous studies have shown that 95% of the schools that have implemented the TEI program have experienced an improvement in coexistence and school climate. In addition, class expulsions are reduced by 40%, school absenteeism by 26%, and some variables, such as performance and students’ self-esteem, improve [37,46]. Another more recent study [36] reports a fall in physical assaults by 52.1%, in verbal harassment by 28.8% and in cyberbullying by 28.4%. In addition, cooperation between students was improved by 28.2% and the social integration of students by 18.1%.

However, the scientific evidence of the TEI program has, to date, only been analysed in schools that have implemented the program and in very specific regions. Therefore, we consider it necessary to carry out a comparative study that allows for the observation of the differences in coexistence, bullying and cyberbullying between the centres in which the TEI program is implemented and those where the prevention program is not carried out in order to answer the following question: Are there significant differences in coexistence, bullying and cyberbullying between centres that have the TEI program and centres that have not implemented the TEI program?

Based on the demonstrated benefits of the program and the results of previous research, it is hypothesized that the educational centres that follow the TEI program enjoy better coexistence and lower rates of bullying and cyberbullying. 

With the interest of deepening knowledge on the effectiveness of the TEI program for improving the coexistence of schools and their ability to reduce bullying and cyberbullying, the aim of this research is to achieve the following specific objectives:-To observe the existing and emerging relations between the different factors involved in educational coexistence.-To compare the coexistence relationships among students in educational centres that are implementing the TEI program to those that do not.-To verify the relationships that are established between the main roles involved in bullying and cyberbullying (victims and bullies).-To compare the incidence of bullying and cyberbullying between TEI centres and non-TEI centres.

## 2. Materials and Methods

In order to achieve the aforementioned objectives, transversal, extensive, retrospective, descriptive research has been carried out with an ex post facto approach by means of quantitative methodology. A comparative study was carried out between centres that have applied the TEI program (TEI centres-experimental group) and centres that have not (non-TEI centres–control group).

In order for the TEI program to involve all students in the school and to be able to observe the benefits, a minimum of two years must have elapsed since its implementation. For this reason, the selected TEI centres have been implementing the program for at least two years.

Regarding the variables of the research, those that are the object of study, and therefore constitute the dependent variables, are coexistence, bullying and cyberbullying. The independent variable of the study is the implementation of the TEI program, differentiating TEI centres (experimental group) and non-TEI centres (control group).

### 2.1. Participants

In this research, 1015 secondary students in Spanish schools from four autonomous communities (Valencian Community, Castilla y Leon, Extremadura, and Galicia) participated. In each of these regions, a TEI centre and a non-TEI centre took part, belonging to the same locality and with similar sociodemographic characteristics. In total, 510 students who belong to centres in which the TEI program is developed and 505 students from centres that do not have the TEI program were the subject of our study. 

Regarding the distribution of participants in the autonomous community, 293 are from the Valencian Community, 354 from Extremadura, 169 from Castile and Leon and 199 from Galicia. Table 1 shows in greater detail the number of students from TEI centres and non-TEI centres in each of the autonomous communities.

Regarding gender and academic year, the sample is quite evenly distributed, with the participation of 524 boys and 491 girls. With regard to the educational level, 292 belong to 1st of ESO (generally 12-year-olds), 253 to 2nd of ESO (13-year-olds, 241 to 3rd of ESO (14-year-olds) and 229 to 4th of ESO (15-year-olds). Table 2 details the distribution of boys and girls in each of the participating years.

The sampling used to select the sample was intentional and non-probabilistic, choosing the most representative centres and those that have similar sociodemographic characteristics. For the selection of the centres, it was required that they had at least two years’ experience with the program, as it is the minimum time necessary for the entire educational community to be involved.

### 2.2. Instrument

For data collection, participants were given a questionnaire that included a series of previously validated questions and scales. First, they were asked about different sociodemographic variables (gender, age, academic year, autonomous community and educational centre). This allowed us to know the characteristics of the sample and compare the participants’ responses. To analyse the variables under study, instruments and scales were used that were previously validated for other research projects.

The coexistence and climate of the centre were evaluated with the School Coexistence Scale (ECE) developed and validated by Del Rey et al. [47]. This scale has a high reliability index (α = 0.94) and is composed of 50 items divided into eight factors that reflect the students’ perception of coexistence in the centre:-The Positive Interpersonal Management Factor (α = 0.83) evaluates the interpersonal relationships that exist between teachers, families and students, reflecting the school climate through the communication that is established between these three educational agents.-The victimization factor (α = 0.90) evaluates the exposure of students to negative and violent actions carried out by other students.-The Disruption Factor (α = 0.90) refers to the negative actions carried out by students that interrupt the teaching–learning process.-The Social Network of Peers factor (α = 0.78) analyses the degree of support that exists between peers and the good relationships that are established between them.-The Aggression factor (α = 0.89) evaluates the presence of hostile behaviours that are carried out by students toward their peers.-The Normative Adjustment factor (α = 0.88) analyses the degree of involvement and adherence of students to the norms of the educational centre.-The Indiscipline factor (α = 86) evaluates the actions or behaviours of the students that are contrary to the rules of coexistence of the classroom and centre.-The Teacher Apathy factor (α = 0.92) evaluates the lack of interest, injustice or incoherence of teachers in the management of interpersonal relationships.

Each of these factors was evaluated based on the perceived frequency or actual frequency based on the facts to which it refers, with a Likert scale of 5 points from 1 = never to 5 = always [48].

*The* European Bullying Intervention Project Questionnaire (EBIP-Q), developed by Brighi et al., was used to evaluate bullying. [49]. This scale is composed of 14 items (7 related to victimization and 7 to aggression), valued according to the frequency with which the behaviours raised occur in the participants, using a Likert scale of 5 points, from 0, never, to 4, always [50]. Actions such as hitting, slapping, insulting, threatening, stealing, breaking belongings, excluding, ignoring and spreading false rumours are considered. The internal consistency values of the original test showed a high degree of reliability in its two factors (α victimization = 0.84 and α aggression = 0.73).

In relation to cyberbullying, the Spanish version of the European Cyberbullying Intervention Project Questionnaire (ECIP-Q) [50,51] was applied. This scale consists of two dimensions (cybervictimization and cyberaggression) and enjoys high reliability indexes (α total = 0.87 α cybervictimization = 0.80 α cyberaggression = 0.88). It is composed of 22 items (11 on cybervictimization and 11 on cyberaggression) and is evaluated with a Likert scale of 5 points, where 0 means never and 4 is always. The items refer to saying bad words, threats, impersonation, theft of personal information, and spreading rumours, images and/or personal videos on the Internet.

### 2.3. Procedure and Data Analysis

For the development of the research, an online questionnaire using the Jot-form application was sent to each student by means of a personal link. Prior to their participation, the objectives and purpose of the project were explained to the participants, guaranteeing at all times their anonymity and the confidentiality of their responses. In all cases, informed consent was then obtained. 

The online questionnaire was carried out within the schools during school hours in the presence of the teacher responsible for the class group as well as a member of the research team, who was available to answer all the participants’ questions.

In order to facilitate analysis and interpretation of the results, the data obtained in all scales and dimensions, both in coexistence as well as in bullying and cyberbullying, were transformed to a hundredfold scale, adapting the values to a range of 0–100.

For the analysis of the results, parametric tests were applied, given the high sample size, taking as a representative statistic of the average score of each group (student t and Pearson correlation). For the estimation of the effect size, Cohen’s d statistic was used. The program used to perform the data analysis was the IBM SPSS STATISTICS version 29, and for the elaboration of the graphs, the Excel program.

## 3. Results

The impact of the TEI program in educational centres was analysed through the evaluation of educational coexistence and the incidence of bullying and cyberbullying in TEI centres and non-TEI centres.

### 3.1. Results of Educational Coexistence

Coexistence was evaluated by considering eight factors (Positive Interpersonal Management, Victimization, Disruption, Social Network of Peers, Aggression, Indiscipline and Teacher Apathy) that reflect the coexistence perceived by students in educational centres [48].

#### 3.1.1. Relationship between Educational Coexistence Factors

First, the relationship between the different factors that constitute the Educational Coexistence Scale (ECE) was studied. By means of the Pearson correlation test, it was observed that there is a significant relationship between all factors. 

In the results shown in Table 3, it was possible to verify that the factors may be organized into two groups. The factors which have a more positive implication for educational coexistence (positive interpersonal management, peer social network and regulatory adjustment) correlate between them in a positive and significant way. However, these three factors correlate inversely and significantly with factors that have a negative connotation for coexistence (victimization, disruption, aggression, indiscipline and teacher apathy).

In the second group are the factors that are detrimental to educational coexistence (victimization, disruption, aggression, indiscipline and teacher apathy). These four factors positively and significantly correlate with each other. However, they show negative and significant correlations with the factors that represent a benefit for educational coexistence, showing inverse relationships.

#### 3.1.2. Difference between TEI Centres and Non-TEI Centres in Educational Coexistence

The differences that exist in the different factors of educational coexistence between the centres that are following the TEI program and those that are not were analysed.

The results of Figure 1 show that TEI centres have obtained higher average scores in factors that favourably affect school coexistence, such as regulatory adjustment, a peer social network, and positive interpersonal management. After applying the student *t*-test for independent samples (Table 4), it was possible to verify that in all three factors, these differences are statistically significant (*p* < 0.001).

However, in the factors that have a negative implication for school coexistence (teacher apathy, indiscipline, aggression, disruption and victimization), non-TEI centres obtained a higher average score than TEI centres. After applying the student *t*-test for independent samples, it has been observed that these differences are also statistically significant in the five factors (*p* < 0.001).

### 3.2. School Bullying and Cyberbullying Results

For the calculation of bullying and cyberbullying rates, EBIP-Q and ECIP-Q scales were used, respectively. With both scales, the levels of aggression (bully) and victimization (bullied) were analysed, both in the school environment and in the virtual environment.

#### 3.2.1. Relation of Bullying and Cyberbullying

To observe the relationship between the main protagonists of bullying (victim and bully) and cyberbullying (cybervictim and cyberbully), Pearson’s correlation test was applied. The results (Table 5) show that there is a positive, quite high and significant relationship (*p* < 0.001) between them all. 

These results indicate that there is a tendency for students involved in bullying to assume all roles at the same time, being victims and bullies, both in the physical school environment and in the virtual one. The highest correlation is observed between the roles of cybervictim and cyberbully (r = 0.838; *p* < 0.001), which indicates that on the Internet, people who are victims often act as aggressors as well. In addition, the correlation between bully and cyberbully is also quite high (r = 0.718; *p* < 0.001), indicating that there is a tendency for students who are bullies in the physical educational environment to also be bullies in the virtual environment.

#### 3.2.2. Differences between TEI Centres and Non-TEI Centres in Bullying and Cyberbullying

Regarding the levels of bullying and cyberbullying in TEI centres and non-TEI centres, it can be observed in the graph of Figure 2 that there are lower levels of aggression and victimization in TEI centres, both in the physical face-to-face and in the virtual environment.

In schools where the TEI program is implemented, students identified themselves to a lesser extent as harassed, reporting lower levels of victimization (M = 13.68; SD = 15.98) than in non-TEI centres (M = 16.54; SD = 17.73). Applying the student *t*-test for independent samples (Table 6), we observed that these differences are statistically significant (t(935.258) = −2.676; *p* < 0.01; d = −0.168). In the virtual environment, TEI students (M = 6.56; DT = 10.82) also show lower levels of cybervictimization than students in schools that do not follow the TEI program (M = 9.89; DT = 15.99). These differences are statistically significant, as indicated by the student *t*-test for independent samples (t(1012.998) = −3.950; *p* < 0.001; d = −0.235).

Focusing on the role of the bully, it becomes clear that participants of TEI centres (M = 7.49; DT = 12.30) have reported to a lesser extent that they enact aggressive behaviours directed toward peers than students of non-TEI centres (M = 10.90; DT = 16.29). This same phenomenon is extrapolated to the virtual environment, where students of TEI centres (M = 4.09; DT = 8.38) ensure that they carry out fewer cyberaggression acts than students of non-TEI centres (M = 7.99; DT = 16.07). After applying the student *t*-test for independent samples, we observed that these differences in identification as aggressors are statistically significant, both in the physical world (t(1001.306) = −3.794; *p* < 0.001; d = −0.230) and in the virtual one (t(957.888) = −5.046; *p* < 0.001; d = −0.289).

## 4. Discussion

In educational centres, in addition to training on academic and curricular content, it is necessary to contribute to the personal and social development of students by promoting the values necessary to live in society and favouring a civic coexistence free of violence.

It has been shown that bullying is more frequent in schools and environments with very rigid or arbitrary rules, excessive competitiveness, a lack of support from teachers, or in environments in which there is insufficient awareness of the importance of good coexistence for learning [52]. On the contrary, educational centres that are more involved with and sensitized to the problem of bullying develop preventive actions and programs to prevent this type of behaviour and situations from occurring. 

Therefore, in schools, it is important to develop specific programs such as the TEI program, which seeks to improve the climate and culture of schools, involving the entire educational community while paying special attention to students.

With this need in mind, this research has shown that the TEI program promotes better coexistence in schools and generates lower levels of bullying and cyberbullying when compared to schools where this program is not being undertaken. Thus, the initial hypothesis of our study is confirmed, which is also supported by previous research that has demonstrated scientific evidence which supports the TEI program [36,37].

Focusing on the first research objective, the evaluation of coexistence in schools through the ECE scale [47], has shown the existence of two groups of clearly differentiated factors. Turning to adequate coexistence in schools, we find a group of factors that positively affect this objective-positive interpersonal management, peer social network and regulatory adjustment. These results show the importance of interpersonal relationships that are generated between the different educational stakeholders (students, teachers and families), the supportive relationships between peers and the existence of clear rules and limits shared by the entire educational community. It has been observed that there is a positive and significant correlation between these three factors, demonstrating that they tend to happen simultaneously in schools where good coexistence is perceived. Conversely, victimization, disruption, aggression, indiscipline and teacher apathy are factors which correlate significantly negatively with coexistence in schools. It is clearly demonstrated that the presence of hostile behaviours among students, the exposure of students to violent actions by their peers, the transgression of the norms that interrupt the correct development of the teaching–learning process and the lack of interest and the lack of involvement of teachers, are elements that have a negative impact to coexistence in schools. These factors, which are clearly detrimental to coexistence, also correlate with each other, showing that they tend to arise simultaneously when adequate coexistence is not perceived in schools.

Comparing educational coexistence between TEI and non-TEI schools, according to the second research objective, it has been revealed that the students of the TEI centres also have a more positive perception of this aspect than students of the non-TEI centres. Higher average values have been obtained in those factors favourable for coexistence (normative adjustment, peer social network and positive interpersonal management) in TEI centres. Conversely, higher average values in the factors that negatively affect coexistence (victimization, disruption, aggression, indiscipline and teacher apathy) are found in non-TEI sites. Thus, students have a more favourable perception of coexistence in the centres where the TEI program is implemented than in those centres in which the prevention program of school bullying has not. These results indicate that the TEI program represents an improvement in coexistence in schools, as has already been reported in other previous studies [36,37].

Regarding the variables of bullying and cyberbullying, and addressing our third research objective, the roles of victim and aggressor have been analysed, both in traditional bullying and cyberbullying. The results suggest that there is a high and significant relationship between the four roles (victim, aggressor, cybervictim and cyberaggressor). These results reveal that those students who are victims also act as aggressors, this phenomenon occurring both in traditional harassment and cyberbullying. Thus, it is confirmed that there exists a relationship between the primary roles in school bullying, victims and bullies. These findings had also been demonstrated in previous studies where continuity and overlap between victim and aggressor roles were found, especially in the virtual context [25]. In addition, students who are bullies in the physical environment also tend to be bullies in the virtual environment. Similarly, those students who suffer as victims in schools are also often subjected to attacks and humiliations on the Internet. 

Focusing on the fourth and final research objective, the incidence of bullying and cyberbullying has been compared between TEI and non-TEI schools. It has been shown that TEI centres report fewer aggressors and victims, both in traditional harassment and in cyberbullying. These results show that the TEI program contributes favourably to reducing bullying and cyberbullying, as other studies have previously affirmed [36,37]. Therefore, the results of this study indicate a clear link between the TEI program and the prevention of bullying and cyberbullying, as well as the improvement of educational coexistence.

Previous studies [20,53] have shown that adolescents have a greater ability to cause longer-lasting harm to their victims on the Internet due to their greater scope of dissemination and for the preservation of the anonymity of the aggressor on many occasions. During the months of COVID-19 lockdown and online education, there arose an alarming growth of cyberbullying as a result of confinement and the abnormally frequent use of electronic devices [21,22]. It is, therefore, essential to train students on the risks that the misuse of electronic resources and social networks entails, educating them on digital responsibility and respect for other users. The TEI program includes a module for raising awareness and training students on the phenomenon of cyberbullying and the responsible use of technologies and social networks. It is necessary to continuously update these training initiatives in order to adapt to the constant changes that occur in the use of technologies and in the types of digital violence taking place, incorporating the new platforms and modes of relationship that adolescents use in social networks.

## 5. Conclusions

The presence of bullying and cyberbullying causes innumerable negative consequences for all those involved, which affects the students’ mental and emotional health. There exists scientific evidence that mental disorders are increasing among adolescents, with symptoms such as anxiety, depression, isolation, obsessive routines, concentration span problems, sleep disorders, autolytic disorders and even suicidal tendencies or symptoms of post-traumatic stress. These are all often the result of bullying and cyberbullying [53]. In addition, victims can also become aggressors, reproducing those acts of violence they previously suffered. If we analyse the consequences from the point of view of the aggressors, we can appreciate how they usually do not appear immediately but with the passage of time. They may suffer progressive loss of peer support, rejection, and isolation. This may also give rise to future criminal problems such as street violence or the use of illegal substances [18]. Other authors [13,53] also highlight the increased risk of school dropout, insensitivity to other people’s emotions and difficulty in complying with rules. In the case of witnesses or observers, the most perceivable consequences are feelings of guilt after having remained passive in the face of the conflict or feelings of vulnerability for fear of being the next victim.

As can be seen, bullying not only has negative consequences for the student who suffers or carries it out but for the entire educational community that witnesses it, be it directly or indirectly. It is, therefore, essential to design educational programs or strategies that help prevent bullying and promote coexistence and the creation of a healthy school climate. The TEI program is aimed especially at developing emotional skills and promoting values such as empathy, cooperation, empowerment and commitment in students as preventive and deterrent elements of violent behaviour. These strategies promote psychological well-being, and reduce the appearance and development of mental disorders, which are sadly rife among adolescents in recent times.

It is important to note that this study is based on the collection of data through self-report instruments, which implies certain inherent limitations. The data obtained rely on responses provided by participants, which may be subject to cognitive biases, lack of precision, or even the possibility of socially desirable responses. Additionally, self-assessment can be influenced by personal and subjective factors, which could affect the objectivity of the results. Despite these limitations, measures have been taken to ensure the confidentiality and sincerity of responses, and appropriate statistical techniques have been used to minimize the impact of potential biases. However, it is essential to recognise that the use of self-report instruments represents a methodological constraint that should be taken into account when interpreting the findings of this study.

Another significant limitation of this research has been the sample selection. The number of schools that participated was limited in comparison to the total number of study participants. This limitation implies that the results and conclusions should be interpreted with caution and may not be applicable to a broader population of educational institutions. For future research, it is recommended to consider a more diverse and larger sample of schools to improve the generalizability of the findings. Despite this limitation, this study provides a solid foundation for future research in the field of coexistence and the prevention of school bullying.

Thanks to this research, we have made progress in understanding the effectiveness of the TEI Program in improving educational coexistence and reducing levels of school bullying and cyberbullying, laying the groundwork for future investigations in this field. One possible direction for future research could be the development of longitudinal studies that allow us to track changes in students from the beginning of the TEI program’s implementation and follow the participants’ progress over time. Additionally, comparing our findings with similar research in different geographical or cultural contexts could further enrich our understanding of the program’s effectiveness.

In the future, it is necessary to continue researching the comorbidity that exists between school bullying and mental health while also examining the benefits that may arise from the TEI program for promoting psychological well-being and reducing emotional problems in students. Certainly, it would also be interesting to continue researching the effects of bullying in more advanced academic stages, such as university studies [54], as well as the effectiveness of the TEI Program in that educational stage.

In this research, the TEI Program has emerged as one of the primary, highly effective strategies for promoting positive school coexistence and preventing school bullying and cyberbullying. It has been demonstrated that the advantages of the TEI Program are significant, with students in schools implementing the TEI Program experiencing better educational coexistence and a lower incidence of school bullying and cyberbullying compared to those in schools where the program is not implemented. Furthermore, the TEI Program promotes active student participation by involving older students as mentors for their younger peers. This peer mentoring dynamic contributes to creating a more inclusive and safe school environment where students feel supported and engaged, fostering values such as empathy, respect, and solidarity among peers. Moreover, empowering students as active agents in bullying prevention makes them feel like an essential part of the process, encouraging their involvement in the early identification of conflict situations and peaceful conflict resolution, and thus reducing levels of school bullying and cyberbullying.

In conclusion, the results of this study support the effectiveness of the TEI Program as a valuable tool in the educational realm, as it promotes a collaborative learning environment and mutual support. These findings underscore the importance of considering the implementation of the TEI Program as an effective tool for promoting the improvement of educational coexistence and reducing cases of school bullying and cyberbullying through active student participation in the process.

## Figures and Tables

**Figure 1 ijerph-20-06818-f001:**
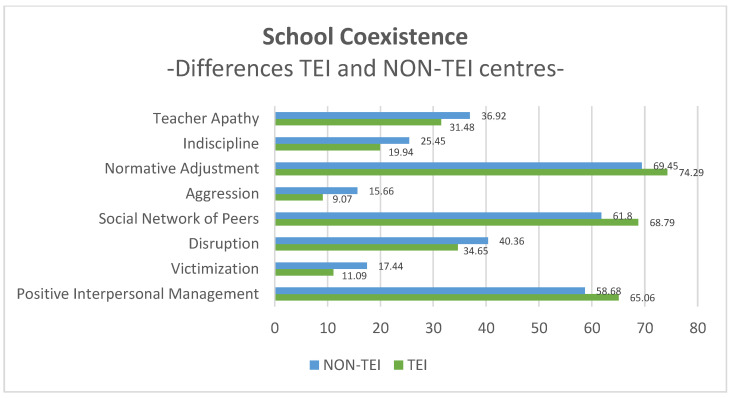
Average scores of TEI centres and NON-TEI centres in factors of School Coexistence.

**Figure 2 ijerph-20-06818-f002:**
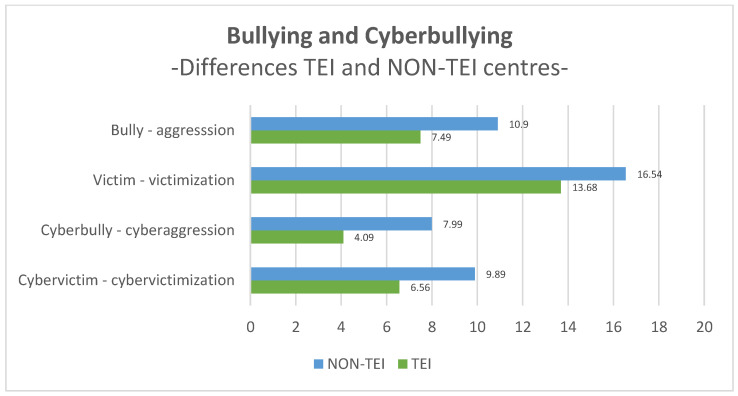
Average scores of TEI centres and NON-TEI centres on Bullying and Cyberbullying.

**Table 1 ijerph-20-06818-t001:** Participants TEI and non-TEI centres by Autonomous Community.

	TEI	NON-TEI	TOTAL
Valencian Community	138	155	293
Extremadura	167	187	354
Castile and Leon	90	79	169
Galicia	115	84	199
TOTAL	510	505	1015

**Table 2 ijerph-20-06818-t002:** Distribution of participants by gender and academic year.

	BOYS	GIRLS	TOTAL
1° ESO	156	136	292
2° ESO	135	118	253
3° ESO	120	121	241
4° ESO	113	116	229
TOTAL	524	491	1015

**Table 3 ijerph-20-06818-t003:** Pearson Correlations between educational coexistence factors.

	Positive Interpersonal Management	Victimization	Disruption	Social Network of Peers	Aggression	Regulatory Adjustment	Indiscipline	Teacher Apathy
Positive interpersonal management	1	−0.244 ***	−0.154 ***	0.537 ***	−0.256 ***	0.577 ***	−0.235 ***	−0.372 ***
Victimization		1	0.432 ***	−0.356 ***	0.572 ***	−0.278 ***	0.346 ***	0.266 ***
Disruption			1	−0.097 **	0.333 ***	−0.142 ***	0.249 ***	0.338 ***
Social Network of Peers				1	−0.213 ***	0.418 ***	−0.148 ***	−0.138 ***
Aggression					1	−0.382 ***	0.491 ***	0.311 ***
Regulatory adjustment						1	−0.436 ***	−0.315 ***
Indiscipline							1	0.367 ***
Teacher Apathy								1

** *p* <0.01; *** *p* <0.001.

**Table 4 ijerph-20-06818-t004:** Test *t* student independent samples in educational coexistence factors between TEI centres and non-TEI centres.

	Gender	N	M	DT	t	gl	p	d
Positive interpersonal management	TEI Centres	510	65.056	18.915	5.141	1013	<0.001	0.329
	Non-TEI Centres	505	58.683	19.686				
Victimization	TEI Centres	510	11.090	14.193	−6.188	996.329	<0.001	−0.377
	Non-TEI Centres	505	17.437	18.411				
Disruption	TEI Centres	510	34.648	17.351	−4.798	960.869	<0.001	−0.298
	Non-TEI Centres	505	40.361	20.330				
Social Network of Peers	TEI Centres	510	68.790	18.383	5.758	1013	<0.001	0.368
	Non-TEI Centres	505	61.803	19.356				
Aggression	TEI Centres	510	9.070	16.299	−5.465	1005.728	<0.001	−0.330
	Non-TEI Centres	505	15.661	22.090				
Regulatory adjustment	TEI Centres	510	74.285	20.167	3.604	1013	<0.001	0.231
	Non-TEI Centres	505	69.452	21.483				
Indiscipline	TEI Centres	510	19.939	23.330	−3.644	1013	<0.001	−0.233
	Non-TEI Centres	505	25.455	23.884				
Teacher Apathy	TEI Centres	510	31.484	30.610	−3.232	1013	<0.001	−0.207
	Non-TEI Centres	505	36.921	22.928				

**Table 5 ijerph-20-06818-t005:** Bivariate correlations between bullying and cyberbullying.

	Victim	Bully	Cybervictim	Cyberbully
Victim	1	0.641 ***	0.643 ***	0.511 ***
Bully		1	0.663 ***	0.718 ***
Cybervictim			1	0.838 ***
Cyberbully				1

****p* < 0.001.

**Table 6 ijerph-20-06818-t006:** Student’s *t* test, bullying and cyberbullying samples between TEI centres and non-TEI centres.

	Gender	N	M	DT	t	gl	p	d
Victim	TEI Centres	510	13.678	15.978	−2.676	935.258	0.008	−0.168
	Non-TEI Centres	505	16.538	17.727				
Bully	TEI Centres	510	7.491	12.300	−3.794	1001.306	<0.001	−0.230
	Non-TEI Centres	505	10.901	16.295				
Cybervictim	TEI Centres	510	6.563	10.816	−3.950	1012.998	<0.001	−0.235
	Non-TEI Centres	505	9.887	15.992				
Cyberbully	TEI Centres	510	4.091	8.376	−5.046	957.888	<0.001	−0.289
	Non-TEI Centres	505	7.994	16.074				

## Data Availability

The data presented in this study are available upon request from the corresponding author. The data are not publicly available due to privacy and ethical restrictions.

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
