# Peer review of "Impact of the TEI Peer Tutoring Program on Coexistence, Bullying and Cyberbullying in Spanish Schools"

_ijerph, 2023, doi:10.3390/ijerph20196818_

Round 1

Reviewer 1 Report

Authors must:

1. The title is appropriate. 

2. The sample used should be adjusted. 8 schools are not representative of the existing reality, unlike the 1015 secondary school students.

3. Remove content from the introduction (lines 112-117) and place it in the discussion.

4. Join text between line 138 and 139, because there is no reason for a different paragraph. There is continuity in the contents.

5. Delete one specific objective (line 208) and merge two objectives into one, because they are very similar.

6. Address the stated objectives in the discussion. 

7. Draw strong conclusions on the advantages of the TEI peer tutoring programme.

Author Response

First and foremost, we would like to express our sincere gratitude to Reviewer 1 for their valuable comments and observations. We deeply appreciate the time and effort invested in reviewing our article. The reviewer's suggestions and insights have proven to be instrumental in enhancing the quality of our work and in providing greater clarity to the conclusions presented in our study.

Below, we outline the changes we have made in accordance with their recommendations:

  1. Regarding the comment concerning the selection of the study sample, we acknowledge the reviewer's observation as a limitation. Unfortunately, at this stage of the study, it is not feasible for us to increase the sample size and the number of participating schools. Additionally, such an expansion would entail working with different time cohorts. For this reason, we have chosen to include this circumstance as one of the inherent limitations of our study in the conclusion section.

  1. Following your recommendation, we have proceeded to remove the content corresponding to lines 112-117 and have relocated it to the second paragraph of the discussion section, in accordance with your suggestions.

  1. The text from lines 138 and 139 has been merged to recognize the continuity of the content presented in those lines.

  1. In order to streamline the specific objectives and avoid duplications, we have chosen to eliminate the first objective, as it was deemed too general and integrated into the subsequent ones. However, we have retained the last two objectives, as we believe they address distinct issues and have been discussed in separate sections of the results. The specific objective mentioned in line 206 has been analysed in the results section 3.2.1, and the specific objective mentioned in line 208 has been addressed in the results section 3.2.2. Moreover, we believe this approach aligns with the description of the specific objectives in lines 202 and 204 related to the coexistence variable. Nonetheless, we remain open to your guidance should you suggest a different approach, such as grouping the specific objectives into two distinct ones, if deemed more appropriate.

  1. In order to ensure greater clarity of the recommendations and in full accordance with your guidance, we have explicitly highlighted the objectives that have been addressed in the study.

 To enhance clarity, and in accordance with your recommendations, we have explicitly indicated throughout the discussion the objectives that have been addressed in the study.

  1. Finally, at the end of the conclusion section, we have added several paragraphs to provide a more robust specification of the advantages of the TEI Program.

We would like to express our gratitude once again for your guidance, and we remain available should any further modifications or improvements to the document be necessary.

Reviewer 2 Report

Impact of the TEI peer tutoring program on coexistence, bullying and cyberbullying in Spanish schools is examined in this research paper. It is an interesting research approach providing useful results. However, there are some points that need to be improved in order to improve the overall quality of the manuscript:

-         - In p.3, line 150, reference (Gonzalez-Bellido, 2021) is missing in the reference list. Citation format also needs correction.

-          - I suggest the use of more sections. The paper will become more easier to read. Related work and program details are all mixed up and they need to be separate. I suggest the authors to use an additional section for the Related work and a Conclusions section for the last paragraphs of the paper.

-         -  Research results are based on self-assessment, posing a limitation to the study. I think that this should be mentioned in the paper.

-          - paper is mainly based on Spanish language references. This poses obstacles for non-Spanish speakers. Therefore, I suggest the use of additional references in English language wherever possible.

Paper is well written.

Author Response

We would like to express our gratitude to Reviewer 2 for their valuable comments and suggestions, as we believe they have significantly contributed to the improvement of our article.

Below, we outline the changes we have implemented in accordance with your recommendations:

  1. The citation from line 150 (González-Bellido, 2021) has been correctly formatted according to the journal's guidelines.

  1. In accordance with your recommendations, we have divided the introductory paragraph into three distinct sections: 1.1. Educational Coexistence; 1.2. Bullying and Cyberbullying in Schools; 1.3. The TEI Program for the Prevention of School Bullying.

Furthermore, we have split the last section of the article into two separate subsections: Discussions and Conclusions.

  1. We have added a paragraph in the conclusions to explain the limitation that the data has been collected through self-report instruments, and therefore, the results are based on self-assessment.

  1. To enhance the accessibility of our research to a more international audience, we have incorporated several additional references in English [25, 29, 30, 31, 43, 44, 47, 53] into our article. We hope that this will make our work more inclusive and comprehensible to readers who are not familiar with Spanish.

We would like to express our gratitude once again for your constructive comments, which we undoubtedly believe have contributed to improving the overall quality and impact of our article.

Round 2

Reviewer 1 Report

The article has been improved and can be published. Just advise the authors to reference the publication:

Gómez-Galán, J.; Lázaro-Pérez, C.; Martínez-López, J.Á. Trajectories of Victimization and Bullying at University: Prevention for a Healthy and Sustainable Educational Environment. Sustainability 202113, 3426. https://doi.org/10.3390/su13063426

In order to reinforce their references

Author Response

Dear Reviewer 1,

Thank you for your valuable feedback and for reviewing our manuscript again. We greatly appreciate your time and effort in helping us improve our work.

We have included the reference you suggested in the revised manuscript as you recommended [55]. We believe that this addition will strengthen the quality and relevance of our work, and we thank you for highlighting this important source.

Once again, thank you for your valuable input, and we look forward to the publication of our article.

We remain available in case it is necessary to make further modifications or improvements to the document.

Sincerely,